# 🦜 PARROT: SEAMLESS SPOKEN DIALOGUE INTERACTION WITH DOUBLE-CHANNEL LARGE LANGUAGE MODELS

## ABSTRACT

Recent advancements in large language models (LLMs) have demonstrated significant potential in enhancing real-time spoken interactions. Presently, open-source methodologies predominantly depend on intermediate generative text-based transcriptions to manage real-time spoken dialogues. However, these techniques often struggle with providing seamless interactions that involve real-time streaming audio inputs. In this research, we unveil an innovative spoken dialogue language model, **Parrot**, distinguished by its unique pre-training and supervised fine-tuning (SFT) pipeline. This pipeline deviates from conventional methodologies by utilizing both single-channel audio data and double-channel spoken dialogue data to train the textless speech language model. During pre-training, we transform single-channel audio input into a sequence of discrete tokens, thereby instructing the LLM to identify audio tokens via next-token predictions. In the SFT phase, we pioneer a novel approach to double-channel generative spoken dialogue language modeling with a unique "next-token-pair prediction" objective, facilitating the LLM's comprehension of natural human conversations. Our pipeline equips LLM to produce spoken interactions that are more natural and fluid than those generated by baseline approaches, as substantiated by thorough evaluations[1].

## 1 INTRODUCTION

The advent of large language models (LLMs), particularly the GPT series (Patel et al., 2023; OpenAI, 2023; 2024), has profoundly transformed the field of artificial intelligence. These powerful language models attain their capabilities through pretraining on extensive text corpora using decoder-only transformer architectures, guided by an autoregressive next-token prediction objective function. Recently, there has been an increasing interest in integrating the LLMs with other modalities, such as images (Radford et al., 2021; Li et al., 2022; 2023; Liu et al., 2023b), audio (Zhang et al., 2023a; 2024a; Hassid et al., 2023), protein sequences (Lin et al., 2022; Madani et al., 2023) and etc . Among these modalities, audio or speech data holds particular importance as it enables LLMs to engage in real-time voice interactions with humans. The recently unveiled GPT-4o model (OpenAI, 2024) exhibits a remarkable proficiency in managing real-time interactions with users in conversational contexts. Throughout the demo presentation, it was able to generate authentic emotional responses and engage users with swift reactions. These functionalities, however, introduce additional challenges, as the model must thoroughly interpret the distinct audio information within human speech while conducting inference with minimal delay.

Presently, the academic community primarily utilizes open-sourced models (Zhang et al., 2023a; Xie & Wu, 2024; Rubenstein et al., 2023; Huang et al., 2024; Wang et al., 2023a; Nachmani et al., 2024; Wang et al., 2023b) following a cascading approach. This method heavily depends on an intermediate text generation step and generally consists of three stages: automatic-speech-recognition (ASR), text-based question answering (Text-QA), and text-to-speech (TTS) synthesis. While this approach is reliable due to the incorporation of powerful text-based LLMs, it does present three significant drawbacks: (1) **Audio Information Loss:** Audio signals, unlike text, include additional

---

[1]Demo and code can be found at `https://anonymous.4open.science/r/Parrot`.

Figure 1: (a) The cascading approach depends on the intermediate text-based response generation translated by ASR and TTS; (b) The encoder-decoder spoken dialogue language modeling encode one of the speaker's audio sequence $Q^a = (q_1^a, q_2^a, ..., q_T^a)$ as condition information to decode another speaker sequence $Q^b$ following the probability distribution $P(Q^b) = \sum_{i=1}^{T} P(q_t^b | q_{t-1}^b, ..., q_1^b, Q^a)$; (c) Our novel decoder-only spoken dialogue language modeling follows the newly proposed next-token-pair prediction paradigm such that $P(Q^a, Q^b) = \sum_{i=1}^{T} P(q_t^a, q_t^b | q_{t-1}^b, ..., q_1^b, q_{t-1}^a, ..., q_1^a)$.

human responses such as laughter, interruptions, pauses, and repetitions, reflecting the speaker's communication style and emotions. The conversion of audio signals to text could potentially result in the loss of this crucial information. (2) **Error Propagation:** The cascading approach consists of three sequential stages. If the initial ASR translation is inaccurate, the subsequent stages will operate on incorrect intermediate data representations. (3) **Real-time Processing Challenges:** In real-world applications, such as the GPT-4o presentation, spoken dialogues require immediate processing. However, incorporating text translation steps inevitably results in a slower process and adds extra latency during inference. Many recently introduced speech LLMs are striving to mitigate these issues. However, they either depend on text generation or remain confined to basic question-answer functions. We will delve into detailed discussions about these approaches in the subsequent related work section and the appendix, given the rapid growth of this research field. Therefore, the aforementioned limitations of cascading approaches highlight the necessity of developing speech-to-speech models capable of managing spoken conversations without the need for text translations.

In this study, we present a novel pre-training and supervised fine-tuning (SFT) pipeline to develop a robust model, referred to as **Parrot**, specifically designed for spoken dialogue language modeling. The pre-training phase begins with the conversion of continuous audio inputs into a sequence of tokens, a process made possible by training a vector-quantized autoencoder (VQVAE) (van den Oord et al., 2017) to reconstruct these audio signals. We then leverage pretrained LLMs as a foundation for continuous learning on *single-channel* audio sequences, with the goal of next-token prediction. This is accomplished by integrating the learned audio tokens into the original text vocabulary. This pretraining stage aids LLMs in capturing the primary latent distribution of audio token sequences. In the subsequent stage, we utilize *double-channel* audio data for SFT. The key advantage is enabling LLMs to directly comprehend how humans engage in natural dialogues. Unlike existing approaches, we introduce a novel "**next-token-pair prediction**" paradigm to model the double-channel spoken dialogue generation using the decoder-only transformer. The comparison between our proposed method and existing techniques are illustrated in Figure 1. We carry out extensive experiments to validate the superiority of our innovative approach. Specifically, **Parrot** consistently outperforms strong baseline methods by 150% and 200% in average in terms of the reflective pause and interruption response accuracy respectively. Additionally, it achieves a low latency of 300ms. In summary, our contributions to the field are as follows:

1) We present a spoken dialogue language model, **Parrot**, featuring an innovative pre-training and SFT pipeline. This novel approach eliminates the need for intermediate text conversions, thereby facilitating more fluid and natural voice interactions with human users at reduced latency.

2) We propose a new paradigm for double-channel spoken language modeling, called next-token-pair prediction, which holds the potential to be readily generalized for autoregressive modeling of multi-channel audio sequence inputs in future explorations.

3) We provide an extensive evaluation of spoken dialogue language models, encompassing several key aspects and metrics for assessing the quality and speed of spoken interactions.

## 2 RELATED WORKS

**Autoregressive Generative Models.** The autoregressive generative modeling has achieved remarkable success in natural language processing, giving rise to a variety of powerful LLMs (Sutskever et al., 2014; OpenAI, 2024; 2023; Patel et al., 2023). Inspired by these LLMs, numerous studies have examined the application of autoregressive modeling in other domains, such as images (van den Oord et al., 2017; Esser et al., 2021; Li et al., 2024; Tian et al., 2024; Lee et al., 2022; Chang et al., 2022), graphs (You et al., 2018), videos (Weissenborn et al., 2020), molecules (Shi et al., 2020; Schwaller et al., 2019) and protein sequences (Madani et al., 2023; Lin et al., 2022). The fundamental concept of autoregressive modeling focuses on iteratively generating the entire segment from the intermediate portion, which is particularly well-suited for the audio generation.

**Multi-modal LLMs.** Multimodal Large Language Models (MM-LLMs) strive to incorporate knowledge from diverse modalities. A key category of MM-LLMs concentrates on developing connectors (Li et al., 2022; 2023; Liu et al., 2023b; Alayrac et al., 2022) that identify knowledge alignment across various modalities. An alternative strategy (Team, 2024; Zhou et al., 2024; Xie et al., 2024) merges all modalities into a cohesive sequence of tokens and utilizes LLMs to sequentially generate them using modified attention masks. These methods (Wu et al., 2024; Su et al., 2023; Fu et al., 2024) even integrate audio as an input modality, and by simply combining text and audio through MM-LLM techniques, they can address one-direction conditional generation tasks such as speech-to-text translation (e.g., ASR and spoken language understanding) (Radford et al., 2023; Zhang et al., 2023b; Deshmukh et al., 2023; Arora et al., 2023; Tang et al., 2024; Chu et al., 2024; Zhou et al., 2023; Ravanelli et al., 2021; Gao et al., 2023) and text-to-speech translation (e.g., TTS) (Elizalde et al., 2023; Liu et al., 2023a; Huang et al., 2023; Nachmani et al., 2023; Yang et al., 2023; Kreuk et al., 2023; Borsos et al., 2023; Copet et al., 2023; Chen et al., 2024; Anastassiou et al., 2024; Jiang et al., 2023b; Kong et al., 2021; Shen et al., 2024; Casanova et al., 2022; Siuzdak, 2024; Yang et al., 2024; Kharitonov et al., 2023; Le et al., 2023). However, these methods are limited to handling multi-turn multi-modal QA tasks (where the model produces an answer only after the question is completed, as signaled by pressing the input button, for instance) and thereby struggle with real-time voice interaction tasks, which is the primary focus of our work.

**Generative Spoken Language Modeling.** The core concept of our approach relies on the pretraining of robust speech foundation models, with language model learning (LLM) serving as a crucial component, to enable rapid adaptation to a broad spectrum of downstream speech tasks. Much of the prior research has utilized the encoder-decoder architecture to enhance pre-training (Borsos et al., 2023; Lakhotia et al., 2021; Kharitonov et al., 2022; Polyak et al., 2021; Chen et al., 2023; 2022; Hsu et al., 2021; Zeghidour et al., 2022; Défossez et al., 2023; Agostinelli et al., 2023; Ao et al., 2022; Tang et al., 2022; Wu et al., 2023). However, this architecture proves inadequate for handling real-time speech interactions with streaming audio inputs, as it requires the encoder to process the entire input simultaneously. In more recent studies, the decoder-only transformer (Maiti et al., 2024; Zhang et al., 2024a; Hassid et al., 2023; Nguyen et al., 2024; Fathullah et al., 2024; Shen et al., 2023; Zhang et al., 2024b; Das et al., 2024) has been employed to model the audio sequence. This approach capitalizes on the potent language capabilities of LLMs while also facilitating the processing of streaming inputs. Motivated by the advent of GPT-4o, newly developed models aim to endow LLMs with speech conversation capabilities (Ma et al., 2024; Zhang et al., 2023a; Xie & Wu, 2024; Rubenstein et al., 2023; Huang et al., 2024; Wang et al., 2023a; Nachmani et al., 2024; Wang et al., 2023b; Défossez et al., 2024). However, these models either rely on text transcriptions or adhere to the aforementioned MM-LLM methods, lacking the ability for natural turn-taking. In stark contrast, our work leverages double-channel spoken dialogue data to directly instruct LLMs in human conversations. A significant contribution in the realm of spoken dialogue language modeling is dGLSM (Nguyen et al., 2023), but it remains confined to the era of using the encoder-decoder architecture. In our research, we elevate the architecture to the most recent decoder-only transformer.

## 3 PARROT: TRAINING AND INFERENCE PIPELINE

Our **Parrot** comprises two essential steps. The first involves pretraining the LLM on single-channel audio token sequences using the traditional "next-token prediction" objective. The second step fine-tunes the LLM on double-channel audio token sequences, employing the innovative "next-token-pair prediction" paradigm. The rationale behind this strategy stems from the fundamental observation that the single-channel audio data can be sourced from the vast amount of open-source data available on the web. However, the primary limitation of single-audio data is its lack of speaker identity information and the overlapping regions between different speakers can be misleading. On the other hand, double-channel spoken dialogue data encapsulates crucial turn-taking events with distinct speaker channels, and any overlapping event can be easily discerned. Nevertheless, the double-channel data necessitates specific pre-processing techniques to segregate the mixed information from the single-channel data. Therefore, it is a naturally inspired strategy to use the large-scale single-channel audio data for pretraining and the moderate-scale double-channel dialogue data for SFT.

### 3.1 AUDIO TOKENIZATION AND SINGLE-CHANNEL AUDIO PRETRAINING

A single-channel audio is a continuous input sequence $\mathbf{x} \in \mathbb{R}^T$ with time length $T$. Owing to the high sampling rate of continuous audio signals, it is essential to employ an audio tokenizer, which extracts valuable features for the purpose of compressing the information. The audio quantizer $\mathcal{Q}$ projects the audio sequence $\mathbf{x}$ into a set of discrete tokens $Q = (q_1, ..., q_{T'}) = \mathcal{Q}(\mathbf{x})$ ($T' \ll T$), where each token $q_t$ is an integer index from the vocabulary $q_t \in [V]$ where the vocabulary size is $V$. We train the audio tokenizer $\mathcal{Q}$ following the VQ-VAE (van den Oord et al., 2017) framework. In contrast to certain prior studies, we directly train the tokenizer on the raw audio signals $\mathbf{x}$, rather than transforming $\mathbf{x}$ into a mel-spectrogram first. We primarily adopt the training strategy presented in SoundStream (Zeghidour et al., 2022), and provide a brief overview of its underlying mechanism.

Specifically, audio inputs $\mathbf{x}$ is fed into an encoder $\mathcal{E}$ to derive down-sampled latent features $\mathbf{f} \in \mathbb{R}^{\frac{T}{r} \times D}$ such that $\mathbf{f} = \mathcal{E}(\mathbf{x})$ with the down-sampling rate $r$ and the latent dimension $D$. This is achieved by the CNN (Krizhevsky et al., 2012) architecture, which can capture the local dependency of $\mathbf{x}$. Then the quantizer $\mathcal{Q}$ converts the latent feature $\mathbf{f}$ to discrete tokens $\mathbf{q} \in \mathbb{R}^{\frac{T}{r}}$ such that $\mathbf{q} = \mathcal{Q}(\mathbf{f})$ where each entry $q_i$ is a quantized integer index. Each latent feature $\mathbf{f}_i$ for time frame $i$ is mapped to the code index $q_i$ of its nearest embedding vector in the Euclidean sense:

$$q_i = \underset{v \in [V]}{\arg\min} \|\mathbf{z}_v - \mathbf{f}_i\|_2, \tag{1}$$

where $\mathbf{z}_i$ denotes the $i$th embedding vector of the learnable codebook $\mathbf{z} \in \mathbb{R}^{V \times D}$ containing $|V|$ vectors. Then the reconstructed audio signals $\hat{\mathbf{x}}$ are obtained through the decoder $\mathcal{G}$ such that $\hat{\mathbf{x}} = \mathcal{G}(\mathbf{z}_q)$ where $\mathbf{z}_q \in \mathbb{R}^{\frac{T}{r} \times D}$ denotes the codebook embedding vectors of the latent feature $\mathbf{f}$ indexed by $\mathbf{q}$. This autoencoder is trained by both the reconstruction loss and discriminator loss through straight-through estimators with stop-gradient operations. We direct readers to (Zeghidour et al., 2022) for a comprehensive description of the architectures and algorithms involved.

After converting the input audio signals into the sequence of audio tokens $Q$, we subsequently supplement these audio tokens into the original LLM's text token vocabulary. Following this, we train the LLMs on the sequence $Q$ using the standard autoregressive approach with the next-token prediction paradigm:

$$p(q_1, q_2, ..., q_{T'}) = \prod_{t=1}^{T'} p(q_t | q_{t-1}, .., q_2, q_1). \tag{2}$$

The next-token prediction loss is calculated by summing the cross-entropy loss, which measures the classification over codebook embedding indices at each time step. Instead of building the audio language model from scratch, we employ Llama 3 as the initial LLM, augmenting its vocabulary with additional audio tokens. While LLMs, after the aforementioned pretraining, can learn the basic audio token distribution, relying solely on single-channel audio data is inadequate for LLMs to effectively comprehend the subtleties of human communication and generate smooth, natural responses. Consequently, we continue to train the speech LLM to learn human speech conversations by utilizing double-channel spoken dialogue data.

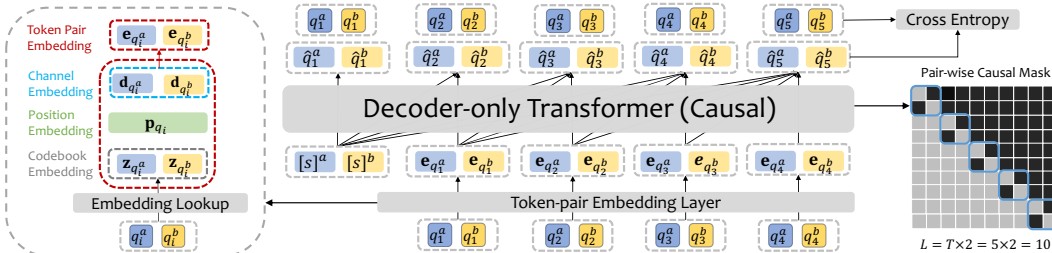

Figure 2: The illustration of the SFT learning mechanism of **Parrot** on the double-channel spoken dialogue data. The novel architecture consists of two important modules. The first module is the embedding layer for obtaining the token-pair embedding; The second module is the decoder-only transformer with a *pair-wise causal masking* attention for next-token-pair prediction. $[s]^a$ and $[s]^b$ denote the special start tokens of channel $a$ and channel $b$ respectively.

## 3.2 SUPERVISED FINE-TUNING WITH DOUBLE-CHANNEL AUDIO

The double-channel audio input comprises a pair of time-aligned single-channel audio inputs, denoted as $(\mathbf{x}^a, \mathbf{x}^b)$, where each channel corresponds to a specific speaker. A fresh challenge arises in the generative modeling of double-channel audio sequences using the decoder-only transformer architecture of LLMs. To address this issue, we propose a novel generative learning paradigm called *next-token-pair prediction*. The key idea here is to generate a sequence of time-aligned token pairs, rather than a single token, in an autoregressive fashion. In contrast to the conventional next-token prediction, our objective is more suitable to the generative modeling of an interpolated dialogue sequence which contain two separate channel identities. Specifically, we begin by discretizing both channels into time-aligned sequences with quantized audio tokens, denoted as $(Q^a = (q_1^a, q_2^a..., q_T^a), Q^b = (q_1^b, q_2^b..., q_T^b))$. To accommodate the input sequence structure within the decoder-only transformer architecture, we reorganize both sequences into a single interpolated dialogue sequence, represented as $Q^{\text{input}} = \{q_1^a, q_1^b, q_2^a, q_2^b, ..., q_T^a, q_T^b\}$. Subsequently, we model the probability distribution that generates the next token pair $(q_t^a, q_t^b)$ at next time step $t$ conditioned on the previously generated token pairs from step 1 to $t - 1$:

$$p(q_1^a, q_1^b, q_2^a, q_2^b..., q_T^a, q_T^b) = \prod_{t=1}^{T} p(q_t^a, q_t^b | q_{t-1}^a, q_{t-1}^b, ..., q_2^a, q_2^b, q_1^a, q_1^b). \tag{3}$$

Then we decompose the token pair conditional generating distribution $p(q_t^a, q_t^b | q_{t-1}^a, q_{t-1}^b, ..., q_1^a, q_1^b)$ by assuming the conditional independence between $q_t^a$ and $q_t^b$:

$$p(q_t^a, q_t^b | q_{t-1}^a, q_{t-1}^b, ..., q_1^a, q_1^b) = p(q_t^a | q_{t-1}^a, q_{t-1}^b, ..., q_1^a, q_1^b)p(q_t^b | q_{t-1}^a, q_{t-1}^b, ..., q_1^a, q_1^b). \tag{4}$$

We illustrate this conditional independence and the dialogue distribution modeling in Figure 1. The probability distribution in Eq.3 and Eq.4 adheres to a fundamental inductive bias that *a person's speech is influenced by both his own previous statements and what he has heard in the past*. To adapt to the generative modeling of the newly arranged dialogue sequence $Q^{\text{input}}$, we need to modify the embedding layer and the attention masking mechanism accordingly. Our novel token-pair embedding layer consists of three important embeddings in total, which are codebook embedding $\mathbf{z}$, position embedding $\mathbf{p}$ and channel embedding $\mathbf{d}$. Specifically, for each token pair $q_t^a, q_t^b$:

$$\mathbf{z}_{q_t^a}, \mathbf{z}_{q_t^b} = \text{lookup}(\mathbf{z}, q_t^a, q_t^b), \quad \mathbf{p}_{q_t^a} = \mathbf{p}_{q_t^b}, \quad \mathbf{d}_{q_t^a}, \mathbf{d}_{q_t^b} = \text{one-hot-embedding}(\mathbf{id}^a, \mathbf{id}^b). \tag{5}$$

In the above Eq. 5, $\mathbf{d}_{q_t} \in \mathbb{R}^D$ denotes the channel embedding of its one-hot identity encoding $\mathbf{id}$, which indicates the speaker role ($a$ or $b$) of token $q_t$. The positional encoding is represented as $\mathbf{p}_{q_t} \in \mathbb{R}^D$ indicating which time step both tokens are from. It is important to note that both $q_t^a$ and $q_t^b$ share the same positional embedding, with the Llama 3 (Dubey et al., 2024) model utilizing the Rotary positional embedding as described in (Su et al., 2024b). After the token-pair embedding layer, we obtain the input embedding $\mathbf{e}_{q_t} = [\mathbf{z}_{q_t}, \mathbf{p}_{q_t}, \mathbf{d}_{q_t}]$ for each token $q_t$ ($a$ or $b$). Following the

implementation of Llama 3, we add both positional embedding and channel embedding to the query and key vectors (instead of value vectors) of each token pair as follows:

$$\mathbf{q} = \mathbf{W}_Q[\mathbf{z}_{q_t^a}, \mathbf{z}_{q_t^b}] + [\mathbf{p}_{q_t^a}, \mathbf{p}_{q_t^b}] + [\mathbf{d}_{q_t^a}, \mathbf{d}_{q_t^b}], \ \mathbf{k} = \mathbf{W}_K[\mathbf{z}_{q_t^a}, \mathbf{z}_{q_t^b}] + [\mathbf{p}_{q_t^a}, \mathbf{p}_{q_t^b}] + [\mathbf{d}_{q_t^a}, \mathbf{d}_{q_t^b}]. \quad (6)$$

Following the above Eq. 6, we obtain the query and key matrices for all token pairs, represented as $\mathbf{Q}, \mathbf{K} \in \mathbb{R}^{2T \times D}$, which are projected by weight matrices $\mathbf{W}_Q, \mathbf{W}_K$ respectively. Then we separately multiply codebook embedding vectors by $\mathbf{W}_V$ to obtain the value matrices $\mathbf{V} \in \mathbb{R}^{2T \times D}$. Based on these vectors, we conduct the attention computation as follows:

$$\mathbf{O} = \text{SoftMax}((\mathbf{Q}\mathbf{K}^T/\sqrt{D}) \cdot \mathbf{M})\mathbf{V}, \ \mathbf{M} \in \mathbb{R}^{2T \times 2T}. \quad (7)$$

The pair-wise causal masking matrix $\mathbf{M} \in \mathbb{R}^{2T \times 2T}$ is used to mask out the entries in the self-attention matrix, preventing each token $q_t$ from attending to future tokens ($q_{t'}, t' > t$) and simultaneously attending to tokens from another channel at the same time (i.e. $q_t^a$ and $q_t^b$ cannot attend to each other). The final layer output embedding, denoted as $\mathbf{O}^l \in \mathbf{R}^{2T \times 2T}$, is utilized to generate the next-token-pair prediction $(\hat{q}_{t+1}^a, \hat{q}_{t+1}^b)$ for each $(q_t^a, q_t^b)$ via classifications over codebook embedding indices. The total training loss is equal to the sum of cross-entropy loss over all generated token pair predictions and the ground-truth token pairs. The overall modified embedding layers and self-attention layers are illustrated in Figure 2. Certain advanced architectural components present in Llama 3, such as grouped-queries attention and feedforward layers, have been omitted here, as our modifications do not impact them.

### 3.3 STREAMING INFERENCE

In order to simulate a real-time user-assistant communication scenario, our speech LLM **Parrot** should be proficient in conducting conditional inference with streaming user voice input. In this inference setting, one speaker's voice input is provided as the user, and the model is assigned the task of inferring the other audio channel. This creates a situation that resembles a constrained generation problem. If the inference process strictly follows the training process, then the model should predict $\hat{q}_t^b$ immediately after receiving the speaker's voice input $q_t^a$ at time $t$. However, due to the VQ-VAE audio tokenization mechanism, it's not feasible to receive just a single audio token from the speaker channel during the streaming inference. This is because the VQ-VAE requires a complete audio signal input within a specific time window. Therefore, unlike the training process, we need to determine when the model should start generating spoken re-

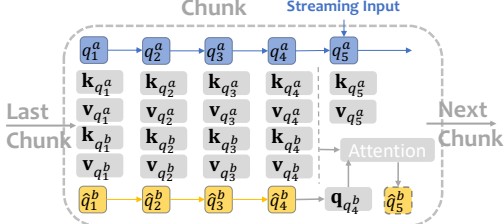

Figure 3: The figure illustrates the chunk-wise streaming inference process. Within each chunk, $(q_1^a, q_2^a, q_3^a, q_4^a, q_5^a)$ represents the provided speaker sequence. Their corresponding keys and values are stored in the KV-cache. **Parrot** sequentially predicts tokens $(\hat{q}_1^b, \hat{q}_2^b, \hat{q}_3^b, \hat{q}_4^b, \hat{q}_5^b)$ based on generated query vectors, which are directed to the Key-Value (KV) cache through attention computations. Once a chunk is filled, the inference process proceeds to the next chunk.

sponses upon receiving streaming user input audio tokens. Specifically, we adopt a divide-and-conquer approach to the inference process, breaking it down into chunks, each containing a pre-determined number of tokens, denoted as $\lambda$. Each time the number of user input tokens reaches $\lambda$ (a chunk of speaker input is given), our model begins to generate predictions until the number of predicted tokens also reaches $\lambda$ (a chunk is filled). This procedure is repeated until the end of user voice inputs (e.g., the conclusion of the voice-assistant service). This inference process is illustrated in the accompanying Figure 3.

## 4 EXPERIMENTS

This section presents the foundational capability evaluation results for **Parrot**. We first describe the two-stage training dataset, data processing methods, and hyper-parameters. We then evaluate the **Parrot**'s performance on core tasks like spoken interaction and provide several case examples.

## 4.1 DATASET

**Parrot** employs a two-stage training process. In the first stage, to establish foundational speech capabilities, we trained the model using three speech datasets totaling approximately 14,000 hours. This stage focuses on both speech understanding and synthesis. Unlike other models (Fang et al., 2024) that require audio to be transcribed into text, our **Parrot** only needs single-channel audio for direct training. This reduces the data requirements and, consequently, increases the amount of training data available. For the second stage, we need the **Parrot** to simultaneously gain the ability to listen and speak. To achieve this, we further utilize the Fisher dataset (Cieri et al., 2004). This dataset comprises 2200 hours of phone conversations between randomly paired participants, each discussing a given topic. A notable feature of the Fisher dataset is that each side of the conversation is recorded on separate channels, which allows us to provide ground-truth separated streams to **Parrot**. The original audio is sampled at 8kHz, and we use Librosa [2] to upsample it to 16kHz.

## 4.2 BASELINES

We compare against baselines from the audio language modeling literature, in three settings. The first category encompasses audio-only models starting from a random initialization, including dGSLM(Nguyen et al., 2023). The second category encompasses several newly released speech LLMs(Zhang et al., 2023a; Xie & Wu, 2024; Fang et al., 2024). As a way to measure the impact of two stage training on spoken fluency, we compare these baselines with **Parrot** trained with and without pre-training phase.

Table 1: The datasets and their usage for training **Parrot**.

| Type | Stages | Dataset | Hours |
|---|---|---|---|
| English Reading speech | 1 | LibriSpeech (Panayotov et al., 2015) | 1,000 h |
| Pronunciation recording | 1 | Common Voice (Ardila et al., 2019) | 3,554 h |
| Video audio | 1 | Gigaspeech (Chen et al., 2021) | 10,000 h |
| Spoken English audio | 1 | Libri-light (Kahn et al., 2020) | 60,000 h |
| Recorded telephone conversation | 2 | Fisher dataset (Cieri et al., 2004) | 2,000 h |
| Speech Instruction | 2 | InstructS2S-200K(Fang et al., 2024) | 100 h |

## 4.3 TRAINING DETAILS

**Large Language Model**: In this study, we conceptualize audio as an additional language and employ three of the most widely recognized open-source LLMs as our foundational models: Llama-3.1-8B(Dubey et al., 2024), Mistral-7B-v0.3(Jiang et al., 2023a), and Gemma-2-9B(Team et al., 2024). Each of these models comprises an embedding layer, multiple transformer blocks, and a language model (LM) head layer. They all encode the relative positional information of tokens using rotary positional encoding (Su et al., 2024a). **Audio Tokenizer**: We train an audio tokenizer based on (van den Oord et al., 2017), which encodes each second of audio into 30-50 discrete tokens from a codebook of size 2048.

## 4.4 PRETRAIN EVALUATION

**Single audio channel language modeling**: We begin by evaluating the capability of **Parrot** to model speech sequences through next-token prediction on the large-scale single channel audio dataset. We use perplexity on the test set's single-channel audio as the metric. The 4a presents the training loss over steps for three distinct models. All three models exhibit a decreasing trend in training loss, indicating effective learning over time. Mistral 7B and Gemma demonstrate similar training loss curves. Notably, Llama 3.1, which exhibits superior text reasoning capabilities, achieves a lower training loss more rapidly compared to Mistral 7B and Gemma. This observation supports our hypothesis that stronger text models can be more effectively adapted to audio tasks, aligning with the conceptualization of "**audio as a new language**".

---

[2]https://librosa.org/doc

We also explore the trade-off between token rate and codebook size to optimize streaming interaction performance in Figure 4c. Notably, the configuration of 30 * 2048, which represents our chosen compromise solution, demonstrates a balanced performance with a steady decline in training loss.

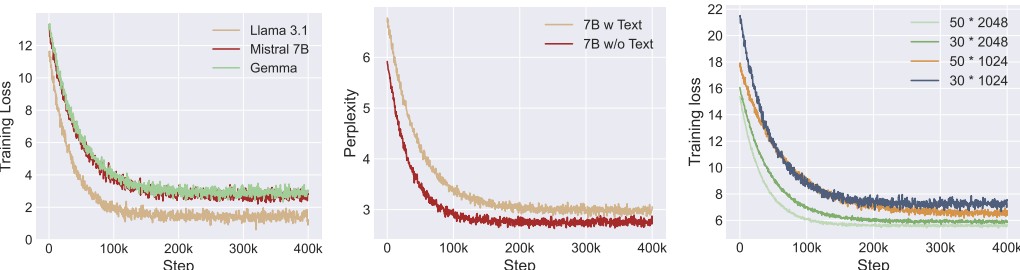

(a) Training loss curve of different foundation models.

(b) The effectiveness of pre-training without text.

(c) The impact of token rate and codebook size.

Figure 4: Training loss and perplexity curves for **Parrot** under various Pretraining settings.

### 4.5 INTERACTIVE EVALUATION

#### 4.5.1 REFLECTIVE PAUSE AND INTERRUPTION EVALUATION

In this section, we use GPT-4 to generate 1,000 text prompts that correspond to two interaction scenarios in natural dialogue: reflective pauses and interruptions. As shown in Figure 5a, reflective pauses evaluate the model's ability to maintain silence during a speaker's contemplative state, while interruptions test the model's response to being interrupted mid-speech. Then we utilize ChatTTS[3] to generate audio prompt corresponding to one of the channels.

Figure 5b compares the accuracy of interaction response by **Parrot** with different baselines. For reflective pauses, **Parrot** demonstrate the highest accuracy at 68%, significantly outperforming the other models. Llama-Omni achieve accuracies of 44%, while SpeechGPT and VITA have an accuracy of only 20% and 19% respectively. Additionally, **Parrot** excelle with an impressive accuracy of 82% for interruption audio prompts, indicating that our model can effectively distinguish human commands.

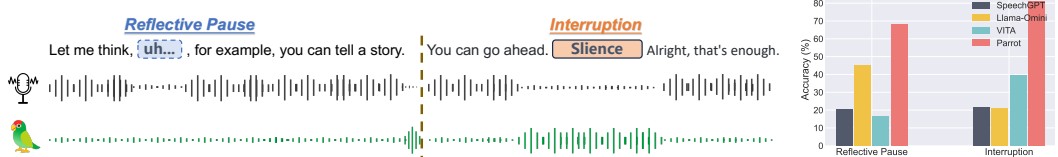

(a) Interactive evaluation settings. Reflective pauses evaluate the model's ability to maintain silence during a speaker's contemplative state, while interruptions test the model's response to being interrupted by the speaker mid-speech.

(b) Interaction turn-taking event response accuracy.

Figure 5: Reflective pause and interruption evaluation.

#### 4.5.2 QUALITY AND STATISTICS OF GENERATED DIALOGUES

We evaluate the linguistic quality and turn-taking dynamics of generated dialogues using various models, as detailed in Table 2. The detailed evaluation settings are in the A.5.2. LSLM(Ma et al., 2024) integrates speaker channels at the embedding layer and separates them in the final layer, demonstrates a notable reduction in the number of Inter-Pausal Units (IPUs) and gaps, indicating smoother transitions between speakers. The dGSLM(Nguyen et al., 2023), particularly with the cross-attention(CA) module, shows a significant decrease in the cumulative duration of pauses and gaps, suggesting more fluid and continuous dialogue. Comparatively, **Parrot** exhibit balanced

---

[3]https://github.com/2noise/ChatTTS

Table 2: Linguistic quality and turn-taking statistics of generated dialogues, including the number of turn-Taking events and cumulative durations per minute, compared to the ground truth.

| Model | Number of occurrences / min | | | | Cumulated duration /min | | | |
|---|---|---|---|---|---|---|---|---|
| | $\Delta$IPU | $\Delta$Pause | $\Delta$Gap | $\Delta$Overlap | $\Delta$IPU | $\Delta$Pause | $\Delta$Gap | $\Delta$Overlap |
| dGSLM w/o CA | -3.9 | 0.9 | -3.6 | -1. | -12.1s | 8.3s | -1.4s | 2.5s |
| dGSLM | -1.6 | 3.4 | -2. | -2.9 | -4.6s | 3.6s | 0.8s | -1.9s |
| LSLM | -2.2 | 3.6 | -2.4 | -3.2 | -4.1s | 3.4s | -1.5s | -2.3s |
| Cascaded | -4.1 | -7. | 7.4 | -6.5 | 1.3s | -5.5s | 0.9s | -3.6s |
| **Parrot$_{0.1}$** | -1.4 | 2.1 | -2.0 | -1. | -3.2s | **2.5s** | -1.2s | -2.1s |
| **Parrot$_{0.5}$** | -1.5 | **1.9** | -1.8 | -1.5 | **-2.9s** | 3.0s | **-0.9s** | -2.2s |
| **Parrot$_{0.9}$** | **-1.3** | 2.2 | **-1.5** | **-0.9** | -3.3s | 2.8s | -1.4s | **-1.9s** |

performance with moderate reductions in both the number and duration of turn-taking events, highlighting their potential for generating natural and coherent dialogues. These findings underscore the importance of model architecture in optimizing dialogue flow and linguistic quality.

## 4.6 HUMAN EVALUATION

We follow the evaluation settings of Veluri et al. (2024) and conduct the evaluation study with 25 annotators with native-level English proficiency. We adapt the Mean Opinion Score (MOS) protocol, utilizing a 5-point Likert scale, to assess the Naturalness (N-MOS) of turn-taking and the Meaningfulness (M-MOS) of dialogue content. Table. 3 compares the meaningfulness and naturalness by **Parrot** with different baselines. Both dGSLM and SyncLLM use Fisher as the only real-world spoken dialogue dataset for training. Besides, we add performance comparison on the out-of-distribution Candor testset(Reece et al., 2023).

Table 3: Meaningfulness (Meaning.) and Naturalness (Nat.) (scores 1-5) mean estimates and standard errors (in parentheses), aggregated overall and for Fisher and CANDOR subsets.

| Model | Overall | | Fisher | | CANDOR | |
|---|---|---|---|---|---|---|
| | Meaning. ↑ | Nat. ↑ | Meaning. ↑ | Nat. ↑ | Meaning. ↑ | Nat. ↑ |
| dGSLM | 1.38 (0.10) | 3.85 (0.12) | 1.82 (0.09) | 4.10 (0.13) | 1.51 (0.12) | 2.85 (0.18) |
| SyncLLM | 3.85 (0.06) | 4.10 (0.05) | 4.10 (0.08) | 4.33 (0.08) | 3.85 (0.09) | 3.91 (0.08) |
| Moshi | 3.90 (0.07) | 3.95 (0.06) | 3.20 (0.10) | 4.32 (0.08) | 3.90 (0.08) | 3.95 (0.08) |
| **Parrot** | 3.95 (0.04) | 4.15 (0.06) | 4.10 (0.06) | 4.42 (0.06) | 4.05 (0.08) | 4.05 (0.10) |
| GT | 4.90 (0.01) | 4.95 (0.02) | 4.90 (0.03) | 4.90 (0.04) | 4.90 (0.02) | 4.95 (0.02) |

## 4.7 ABLATION STUDY

In this section, we present an ablation study to evaluate the impact of different channel embedding designs and training stages on the performance of **Parrot**. The results are illustrated in Figure 6.

**One-stage VS Two-stage**: The Figure 6a compares the perplexity over training steps for these two strategies. The two-stage training approach demonstrates a significantly lower perplexity throughout the training process compared to the one-stage training approach. This indicates that pretraining on single-channel audio data provides a robust foundation, which enhances the model's performance during subsequent fine-tuning on dual-channel data.

**Channel Embedding**: The Figure 6b illustrates the perplexity over training steps for both approaches. The findings indicate that the layer-wise channel embedding consistently achieves lower perplexity compared to the consistent channel embedding. This suggests that enabling each layer to have its own channel embedding allows the model to learn more effective representations, thereby enhancing performance.

In Figure 6c, we present a t-SNE visualization of token embeddings for each channel, derived from the Fisher testing set conversation. Due to the design of the channel embeddings, there is some separation between the token embeddings from different channels in the reduced-dimensional space

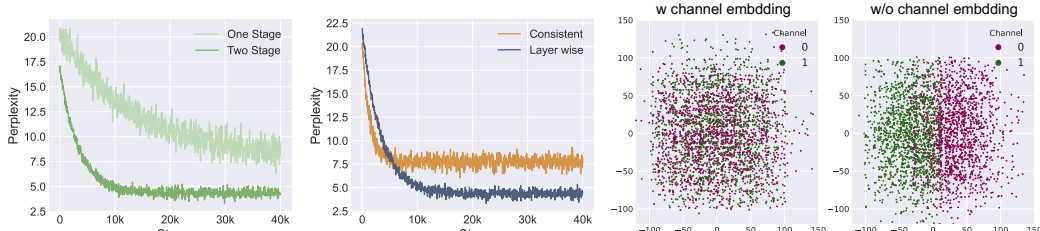

(a) Comparison of perplexity in one-stage and two-stage training strategies.

(b) The impact of channel embedding design.

(c) t-SNE visualization of different channel token embeddings illustrating the distribution of Human (Purple), and Model (Green).

Figure 6: Ablation study on channel embedding designs and training stages.

created by t-SNE. Although there is some overlap between the two channels, these initial findings warrant further exploration and analysis of the embeddings.

## 5 CASE STUDY

```
Scenario: A user engages in a conversation with Parrot, describing an
    object and asking the model to identify it.
User: Please listen to my description of an object below, and say its
    name when you have guessed it. The description is: it has four legs,
    a flat surface, and is often used for dining or working...
Parrot: I guess it might be a table.
```

Figure 7: Case study of **Parrot** interrupt human speaking correctly and timely.

To intuitively understand the differences in responses from our models, we provide an example in Figure 7. In this scenario, **Parrot** interrupts the user at the precise moment it has gathered enough information to make an accurate prediction. This capability is a significant departure from current models that would typically wait for the user to finish speaking before responding. The ability to interject appropriately not only demonstrates the model's advanced comprehension skills but also enhances the fluidity and naturalness of the interaction.

## 6 LIMITATIONS AND FUTURE WORKS

A current limitation of **Parrot** is its incapacity to integrate the prevalent audio tokenization method, residual vector quantization (RVQ) (Lee et al., 2022). RVQ is typically used to convert continuous audio into discrete tokens, ensuring the preservation of high-quality information. This process involves approximating the audio input with multi-scale tokens, each representing the residual information remaining after the deduction of the previous scale token's information. As a result, the audio token sequence produced by RVQ has an additional residual token dimension (beyond the time step dimension) compared to the standard VQVAE (van den Oord et al., 2017) utilized in **Parrot**. This introduces complexities to the autoregressive generative modeling of spoken dialogue sequences.

## 7 CONCLUSION

In conclusion, we introduce a novel spoken dialogue language model, **Parrot**, realized through an innovative pretraining and SFT pipeline. We employ single-channel audio data for pretraining and double-channel audio dialogue data for SFT. To facilitate language modeling on double-channel audio sequences, we unveil the pioneering next-token-pair prediction paradigm for the first time. Comprehensive experiments underscore the superiority of our approach over existing baseline methods. Furthermore, through meticulous ablation studies, we validate the effectiveness of each critical component in our model.

## 8 ETHICS AND REPRODUCIBILITY STATEMENT

In this study, we propose an innovative spoken dialogue language model, **Parrot**. However, it is important to note that we have not yet conducted a comprehensive safety evaluation of this model. While preliminary results are promising, the potential for unintended consequences, such as biases in audio reasoning or misuse of the technology, remains unassessed. We strongly advocate for further rigorous safety and ethical evaluations to be undertaken by the research community to ensure responsible deployment and to mitigate any adverse impacts.

To ensure the reproducibility of our results, we have made our codebase publicly available through an anonymous git repository, which is provided in the footnote of the abstract. This repository contains comprehensive documentation on data processing, model training, and evaluation procedures, as well as the demo display to facilitate understanding and verification of our methods.

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

## A  APPENDIX

### A.1  DETAILED RELATED WORK DISCUSSIONS

We compare **Parrot** with several newly released speech LLMs, which are Mini-Omni (Xie & Wu, 2024), Llama-Omni (Fang et al., 2024), Moshi (Défossez et al., 2024), LSLM (Ma et al., 2024).

1) Mini-Omni: The major advancement of this model is the batched parallel decoding strategy.

- Advantages: Text generation can significantly enhance the quality of the audio produced. Concurrently, the implementation of batched parallel decoding can substantially mitigate issues related to inference latency. Overall, Mini-Omni effectively maintains a high standard of response quality while circumventing the latency typically associated with TTS translations.

- Limitations: This model, while a multi-modal QA system, adheres to the standard architecture of multi-modal LLMs with various modality adaptors. However, it falls short in handling natural spoken conversations with real-time streaming user voice inputs. The dynamic nature of real-time dialogues, characterized by various pauses and turn-taking events, cannot be effectively simulated by this system.

2) Llama-Omni: This speech LLM also mainly focuses on enhancing the decoder stage like the previous Mini-Omni model. It propose an non-autoregressive decoder to simultaneously generate texts and audios. The text token is firstly upsampled and then fed into the speech decoder to derive the output voice. Unlike traditional TTS, Llama-Omni applies TTS word by word in an non-autoregressive manner.

- Advantages: Like the Mini-Omni, this model also enjoys the response reliability due to the usage of intermediate text generation. In this way, Llama-Omni also enjoys low inference latency while maintaining high-quality content response.

- Limitations: The Llama-Omni also shares the same limitations like Mini-Omin. Relying on text generations cannot handle special speech tokens that are hard to match to text tokens. In addition, the multi-modal LLMs can only handle multi-turn QA while failing to handle natural conversations like interruptions and pauses.

3) LSLM: This speech LLM explicitly leverages the double-channel audio data. Unlike **Parrot**, LSLM fuses two channel tokens into one single token and still follows the next-token prediction training objective. To enable LSLM to learn to interrupt, this work trains the speech LLM on the synthetic interruption data.

- Advantages: No need to change the next-token prediction paradigm of the original LLM, which keeps the speech LLM as simple as possible.

- Limitations: The introduction of the special "EOS" token and the "interruption" token will bring additional challenges in audio preprocessing. A threshold must be determined to filter what tokens are assigned to be "interruption token", which can be tricky. In addition, this model can only learn to interrupt by training on specific synthetic data. First, it might be troublesome to synthesize turn-taking events. Second, there is always a distribution gap between synthetic turn-taking and real-world turn-taking.

4) Moshi: This is a newly open-sourced speech LLM with high-quality spoken responses and minimal inference latency. Moshi leverages the RVQ technique to tokenize the audio inputs. And it explicitly proposes the usage of multi-channel audio modeling. There are mainly text channels, speaker audio channels and listener audio channels. The generative modeling of the multi-channel token sequences is following the RQtransformer (Lee et al., 2022), which is an encoder-decoder architecture.

- Advantages: The usage of RVQ can largely improve the quality of discrete audio representations. And the usage of intermediate text translation can significantly improve the reliability of response contents.

- Limitations: The multi-channel data structure requires the alignment between text sequences and audio sequences, which is a non-trivial engineering work. Also, the encoder-decoder RQtransformer architecture requires to receive the entire input of speaker's channel, which still somehow downgrades the modeling efficency. Last but not least, this model can be regarded as alternative form of online cascading approach, which relies on the accuracy of both audio-to-text and text-to-audio generation.

In comparison to the above models, **Parrot** enjoys several important advantages:

- **Real Streaming Inference: Parrot** is capable of managing real-time streaming inference, eliminating the need for specific training on turn-taking, as required by models like LSLM. It can interact seamlessly with human users through natural turn-taking for the duration of

the service. In contrast, multi-modal speech LLMs such as Mini-Omni and Llama-Omni can only interact with users on a turn-by-turn basis. In essence, **Parrot** does not depend on manually-defined interruption rules when conducting streaming inference.

- **Decoder-only Transformers:** In contrast to the encoder-decoder dialogue language modeling, **Parrot** employs a decoder-only transformer. This architecture offers numerous significant advantages. For instance, the encoder-decoder structure necessitates maintaining a window to receive complete inputs during the inference stage. However, the decoder-only architecture simply requires querying the cached key-value pairs, resulting in superior computational efficiency during inference.

- **Spoken Dialogue Data Usage Efficiency:** Both Moshi and LSLM randomly assign one channel as the speaker and another as the listener. This approach potentially reduces dialogue data efficiency, as the trained model becomes speaker-dependent. Essentially, the model needs to train the reverse conditional distribution by swapping the roles, which could pose scalability issues as more channels are added in the future. In contrast, **Parrot** is speaker-independent and concurrently learns the conditional distribution of both speaker's audio channels.

## A.2 REPRESENTATION OF THE JOINT SEQUENCE AND MASK STRATEGY MODELED BY **PARROT**

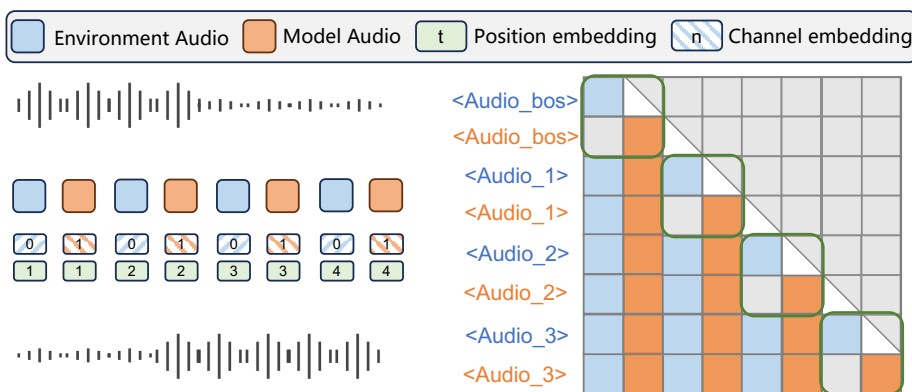

Figure 8: Pair-wise causal masking attention for next-token-pair prediction.

## A.3 POTENTIAL SOLUTIONS TO LIMITATIONS OF **PARROT**

To overcome the limitations previously discussed, we propose a potential solution: the creation of a novel generative model for RVQ-based dual-channel audio sequences. However, the complexity of this task is heightened due to the unclear dependency relations across two distinct dimensions - the time dimension and the residual token dimension. As an alternative, we could opt to refine our method by increasing the number of discrete tokens per second. This approach would circumvent the need for RVQ while simultaneously enhancing the quality of the audio information. In future research, our goal is to train our method on substantially larger datasets and concurrently develop more sophisticated speech language model architectures. We hypothesize that the performance of our method can be further elevated to a new level through various potential approaches, without the direct application of RVQ.

## A.4 MORE IMPLEMENTATION DETAILS AND HYPER-PARAMETER SETTINGS

### A.4.1 HYPER-PARAMETER SETTINGS

Our model is trained on 16 A100 GPUs, utilizing a cosine annealing learning rate scheduler with a minimum learning rate of 4e-6 and a maximum learning rate of 4e-4. Each training epoch consists of 40,000 steps, with batch size 192 for each step. During fine-tuning, we use learn rate from 4e-6 to 5e-5.

### A.4.2 STREAM INFERENCE

Table 4: Latency, speech-text alignment and speech quality under different unit chunk sizes.

| Chunk Size $\Omega$ | Latency (ms) | #Lagging Word | ASR-WER ↓ | ASR-CER ↓ |
|---|---|---|---|---|
| 10 | 310 | 2.1 | 12.5 | 7.42 |
| 20 | 320 | 3.1 | 12.65 | 7.45 |
| 40 | 350 | 4.4 | 12.45 | 7.89 |
| 60 | 410 | 6.9 | 13.10 | 8.10 |
| 80 | 490 | 10.2 | 14.50 | 8.35 |
| 100 | 550 | 11.3 | 15.30 | 9.05 |

## A.5 MORE EXPERIMENTAL RESULTS

### A.5.1 AUDIO TOKENIZER QUALITY

Table 5: Comparison of different models and tokenizers on objective and subjective metrics.

| Model | Tokenizer | Objective | | Subjective | |
|---|---|---|---|---|---|
| | | WER↓ | SIM↑ | MOS↑ | SMOS↑ |
| Groundtruth | | 1.9 | 0.93 | 4.5 | 3.96 |
| VALL-E | EnCodec | 7.9 | 0.75 | 3.08 | 3.31 |
| USLM | SpeechTokenizer | 7.2 | 0.81 | 3.63 | 3.45 |
| **Parrot** | VQVAE | **6.9** | **0.82** | **3.71** | **4.50** |

### A.5.2 DIALOGUE LINGUISTIC QUALITY

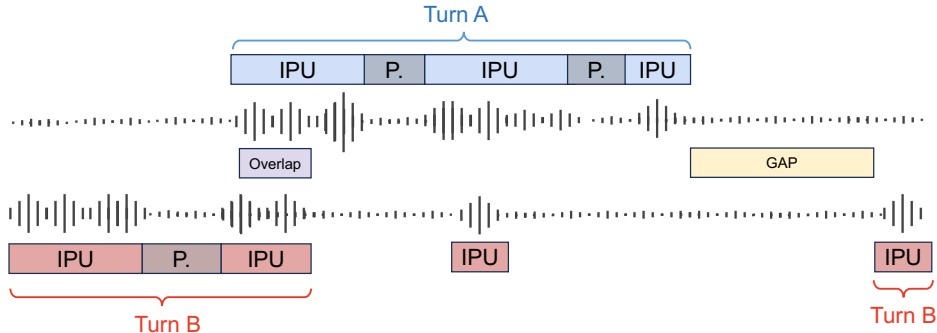

Figure 9: Illustration of turn-taking events: IPU (Interpausal Unit), Turn (for speaker A and Speaker B, resp), P.(within-speaker Pause), Gap and Overlap.

Our model generates two audio channels at the same time, allowing us to use basic Voice Activity Detection (VAD) tools on the output to gather turn-taking metrics. According to the settings in (Nguyen et al., 2023), an Inter-Pausal Unit (IPU) is a continuous speech segment within one speaker's channel, bordered by VAD-detected silences longer than 200ms on both ends. Silence is defined as the lack of voice signals on either channel, while overlap refers to segments where voice signals are detected on both channels. Silences can be further divided into gaps (between IPUs of different speakers) and pauses (within the same speaker's IPUs). Consecutive IPUs by the same speaker, separated by a pause, are merged into a single turn. Our analysis will focus on measuring the duration distribution of IPUs, gaps, pauses, and overlaps in both the training corpus and the dialogues generated by our various models.

### A.5.3 REFLECTIVE PAUSE AUDIO DATASET

---

**Prompt for reflective pause**

"Hmm..., this question is a bit complicated, I need to think about it."
"Let me recall, uh..., yes, we went to the park that day."
"You know, that..., oh, yes, it's the new restaurant."
"I remember he mentioned it, um..., it seems to be last Friday."
"This matter, um..., I think we need to discuss it again."
"Let me think about it, uh..., yes, that's it."
"I'm not sure, um..., maybe I need to confirm it again."
"This question, um..., I think we can solve it this way."
"Let me think about it again, uh..., yes, I remember it."
"The one you mentioned, um..., I seem to have some impression."
"We need to deal with the budget issue of this project. Um..., this problem is a bit complicated, I need to think about it."
"Do you remember the last time we met? Let me recall, uh..., yes, we went to the park that day."
"Have you heard about the new restaurant? You know, that..., oh, yes, that new restaurant."
"When did he tell you the news? I remember he mentioned it, uh..., it seems to be last Friday."
"Do you have any suggestions about this plan? This matter, uh..., I think we need to discuss it again."
"Can you give me an example? Let me think about it, uh..., yes, that's it."
"Are you sure this data is correct? I'm not sure, uh..., I may need to confirm it again."
"How should we deal with this emergency? This problem, uh..., I think we can solve it this way."
"Can you explain this concept again? Let me think about it again, uh..., yes, I remember it."
"Do you know what he is talking about? The one you said, uh..., I seem to have some impression."

---

**Prompt for GPT score**

Content (1-5 points):
1 point: The response is largely irrelevant, incorrect, or fails to address the user's query. It may be off-topic or provide incorrect information.
2 points: The response is somewhat relevant but lacks accuracy or completeness. It may only partially answer the user's question or include extraneous information.
3 points: The response is relevant and mostly accurate, but it may lack conciseness or include unnecessary details that don't contribute to the main point.
4 points: The response is relevant, accurate, and concise, providing a clear answer to the user's question without unnecessary elaboration.
5 points: The response is exceptionally relevant, accurate, and to the point. It directly addresses the user's query in a highly effective and efficient manner, providing exactly the information needed.

Style (1-5 points):
1 point: The response is poorly suited for speech interaction, possibly including structured elements like lists or being overly complex, disjointed, or difficult to understand.
2 points: The response is somewhat suitable but may be too long, too short, or awkwardly phrased, making it less effective in a speech interaction context.
3 points: The response is generally suitable for speech interaction, but it may have minor issues with length, clarity, or fluency that detract slightly from the overall effectiveness.
4 points: The response is well-suited for speech interaction, with appropriate length, clear language, and a natural flow. It is easy to understand when spoken aloud.
5 points: The response is perfectly suited for speech interaction. It is the ideal length, highly clear, and flows naturally, making it easy to follow and understand when spoken.

Below are the transcription of user's instruction and models' response:
### [Instruction]: **{instruction}**
### [Response]: **{response}**

After evaluating, please output the scores in JSON format: {"content": content score, "style": style score}. You don't need to provide any explanations.

---

## A.6 MOTIVATIONS OF USING DOUBLE-CHANNEL SPOKEN DIALOGUE DATA

Inspired by GPT-4o (OpenAI, 2024), we aspire to create a powerful voice assistant that can engage with human users in a natural and fluent way. Ideally, the assistant should be able to be interrupted

by users. If a user needs to convey something urgently, the assistant should stop speaking and listen attentively. Furthermore, when a user is in thought or taking a pause, the assistant should not prematurely conclude that the user has finished speaking. Instead, it should patiently wait for the user to complete their thoughts. An advanced voice assistant could even interrupt users when it has already grasped their intentions, much like how we often interrupt each other in daily conversations. There are numerous other scenarios that an intelligent voice assistant should be equipped to handle. Given these complex application scenarios, it's challenging to address these issues through simple manual engineering, such as the introduction of special tokens like silence tokens, or hard interruptions when the user is speaking.

The success of foundational models hinges on our trust in the model's capacity to learn autonomously from data, rather than over-interfering with the learning process or over-engineering the neural architectures and algorithms. Consequently, in this paper, we utilize double-channel dialogue data and directly train the speech LLM on this spoken dialogue data. With robust pre-trained speech LLMs, we can reasonably anticipate that the model can learn how humans converse with each other by directly "reading" their dialogues. This approach eliminates the need for setting manual rules to assist the voice assistant in scenario judgement. The assistant may learn how to navigate these scenarios by processing a sufficient amount of spoken dialogue data. Regrettably, the current availability of open-source double-channel spoken-dialogue data is limited. Looking ahead, we hope our work will stimulate the community to gather large-scale double-channel or even multi-channel spoken dialogue data.

