# OpenReview forum: "Parrot: Seamless Spoken Dialogue Interaction with Double-Channel Large Language Models"
_ICLR.cc/2025/Conference — Submitted to ICLR 2025_

### Official Review · Reviewer_2fHe · 2024-10-24

**Soundness:** 1
**Presentation:** 1
**Contribution:** 1
**Rating:** 1
**Confidence:** 4

**Summary:**

This work presents an approach to spoken dialogue interaction through the development of a large language model (LLM) named Parrot. The authors propose a two-stage training pipeline that leverages single-channel audio data for pre-training and double-channel audio data for supervised fine-tuning (SFT). The key innovation lies in the "next-token-pair prediction" paradigm, which aims to enhance the model's ability to comprehend and generate natural human conversations in real-time.

**Strengths:**

1. This work explores an interesting topic, namely dual-channel speech input modeling. Previous work has focused on user-assistant turn-based interactions, but in real interactions, immediate processing is required. Therefore, there is a need for a listening channel to continuously process user's speech input.

2. The author made reasonable reviews and citations to related work.

**Weaknesses:**

1. The writing of this paper is very poor, with some tables and figures not being referenced in the main text, and inconsistent statements in the context, making the entire article very difficult to understand. (see Questions for details)

2. The structure design of the model and "next-token-pair prediction" paradigm are not well-motivated, as there is a significant gap between training and inference stages.

    -  The author inputs listening tokens and speaking tokens as pairs into the LLM, which doubles the context length. LSLM [1] has demonstrated the effectiveness of modeling double-channel through embedding fusion and is more efficient in context. Therefore, the author needs to prove the effectiveness of their modeling approach
    -  In the training process, the prediction of the next token depends on the previous predicted tokens. But in the inference process, according to section 3.3, listening tokens are sent to LLM in chunks. At this point, predicting the next speaking tokens no longer depends on the previous predicted speaking tokens. This inference method will disrupt the casual modeling during training.

3. The lack of evaluation details.

      -  In section 4.5.1, LLAMA-Omini and SpeechGPT do not have the ability to interrupt and pause. How does Reflective Pause and Interruption caculated?

      -  In section 4.5.2, all these evaluation metrics cannot reflect the linguistic quality of the model. For an open-ended instruction-following task, there is no evaluation of response quality.

      -  In section 4.6, the evaluation of speech-to-text tasks, such as ASR and ST, is discussed. However, this model is a textless speech-to-speech model and does not have text generation capability, so it is unclear how these tasks are evaluated.

4. Some results are very strange, which it‘s hard for me to believe that it's real.

      -  According to table 1, the model is only trained on English-Only speech data, but in figure 8(c), the model can perform Chinese ASR task on aishell, SER task on MELD, and sound-related task on AIRBench-Sound. This is very strange.
      -  In appendix A.4.2, the larger latency, the worse performance. There is no explanation for this strange results.

**Questions:**

1. In section 4.1, the authors claim to have used 14,000 hours of single-channel for pretraining and 2,200 hours of double-channel for sft. However, in table 1, there are over 70,000 hours of single-channel data and 2,000 hours of double-channel data. At the same time, InstructS2S-200K is a single-channel data, how do you use it in stage 2?

2. In section 4.5.3, the author claims that the embedding of two channels is in different spaces, but in Figure 7, there are two completely different feature spaces, with the right side appearing consistent with the author's statement, so what is the left side of the figure?

3. In Appendix A.6.3, the author lists "prompt for gpt score," but there is no mention in the main text of the need for gpt evaluation.

4. How much data was used to train the speech tokenizer? And what's the output rate of the speech tokenizer?

---

### Official Review · Reviewer_uBhb · 2024-10-28

**Soundness:** 2
**Presentation:** 1
**Contribution:** 2
**Rating:** 3
**Confidence:** 4

**Summary:**

The Parrot framework aims to address double-channel spoken dialogue modeling by implementing a pipeline that includes pre-training on single-channel audio and fine-tuning on double-channel audio data. The framework introduces a "next-token-pair prediction" approach within a decoder-only model architecture. However, the proposed solution lacks substantial originality and leaves questionable details for evaluation, which ultimately weakens the paper.

**Strengths:**

The framework presents a relevant approach to double-channel spoken dialogue, aiming to improve latency by avoiding text generation stages, which could contribute to real-time applications.

**Weaknesses:**

* Minimal Novelty: The framework essentially serves as a decoder-only adaptation of dGSLM combined with a textually pre-trained model (TWIST), offering limited technical innovation. This incremental change does not justify the need for a separate model or paper.

* No Human Evaluation: The absence of human assessment significantly limits the validity of the framework's claims about improving conversational fluidity and natural interaction, which are central to spoken dialogue applications.

* Lack of Established Benchmark Comparisons: Despite the existence of standard benchmarks like ZeroSpeech [1] and StoryCloze [2] for textless spoken language models, the paper does not include comparisons with these datasets. This omission raises concerns about the thoroughness of the experimental validation.

* Poorly Defined Evaluation Methodology: The evaluation details, especially for the reflective pause and interruption response accuracy (Section 4.5.1), are incomplete. Key information, such as the evaluation metric definitions like interaction accuracy, is missing, making it hard to verify the claimed improvements.

* Insufficient Explanation of Key Evaluation Components: See the comments below.

1. Zerospeech 2021 benchmark, https://arxiv.org/abs/2011.11588
2. StoryCloze, https://arxiv.org/abs/2305.13009

**Questions:**

* L473-476: The discussion on "layer-wise" and "consistent" channel embeddings is unclear. These terms appear only once and lack explanation, leaving their meanings and relevance ambiguous to the reader.
* L1162-1187: The purpose of the GPT score is not clear. It is mentioned in the appendix, but its usage and significance are not explained in the main text, making it difficult to understand its role in the evaluation.
8 483-505: Since AIR-bench questions are in text format, it is unclear how the proposed model, which is audio-based, handles text input. Without further clarification, it is difficult to interpret the evaluation results accurately.
* Section A.1: The essential distinctions between the proposed method and closely related prior work should be concisely summarized in the main related works section. The current related works section lacks sufficient comparison, especially with the most relevant prior methods.

---

### Official Review · Reviewer_w4cg · 2024-11-03

**Soundness:** 2
**Presentation:** 2
**Contribution:** 2
**Rating:** 3
**Confidence:** 5

**Summary:**

The paper introduces a spoken dialogue model, Parrot, which leverages large-scale single-channel audio data for pre-training and moderate-scale dual-channel dialogue data for supervised fine-tuning. The authors also propose a “next token-pair prediction” approach for spoken dialogue language modeling. The study claims that Parrot facilitates more natural and fluid conversations compared to Dialog GSLM and traditional cascaded spoken dialogue systems.

**Strengths:**

1. The authors state that they will open-source their training and inference framework, which would be a valuable contribution to the community, especially since existing end-to-end spoken dialogue models either do not disclose their training methodologies (e.g., Moshi) or lack sufficient documentation (e.g., Mini-OMNI).
2. The authors also evaluate their approach on turn-taking properties, such as recognizing when the user has paused and determining appropriate moments to interrupt the user.

**Weaknesses:**

1. The paper lacks specific details on the cascaded system used for evaluation. DialogGSLM used a relatively weak cascaded system in their work, so it would strengthen this study if the authors evaluated a cascaded system with state-of-the-art ASR, LLM, and TTS models, such as Hugging Face's Speech-to-Speech (https://github.com/huggingface/speech-to-speech), for a fairer comparison. Additionally, latency in cascaded systems can be minimized by parallel threading each module, as demonstrated in the Hugging Face repository.

2. Given the primary objective of this work is to enable engaging and naturally fluid conversations, a human study would be valuable for a thorough evaluation—similar to assessments in text-based dialogue systems. When motivating their approach, the authors point out limitations in cascaded methods; more analysis is needed to clarify if these limitations genuinely impact user experience in human-AI conversations. User study could focus on user satisfaction compared to baseline systems as well as other auxiliary metrics such as naturalness of turn-taking and semantic coherence of response.

3. Although the code is public, it lacks sufficient documentation, and I was unable to run their demo. For example, an inference.py file appears to be missing.

4. The paper would benefit from further discussion on design choices:

a. I was curious about the choice to use a single codebook. Table 4’s synthesized audio quality results lack clarity on the dataset used. Multiple codebooks often improve results—did the authors experiment with this option?

b. The motivation for using next-token pair prediction, rather than multi-stream prediction as seen in Moshi, is also unclear. While streaming audio generation is effective, the paper doesn’t address the length limits for audio modeling. Given the potential length of audio sequences, does the model implement techniques to reduce sequence length?

c. The authors use only human-human conversation data for supervised fine-tuning. Adding human-AI conversation data, even synthetic, could be beneficial as there are nuance differences in how humans communicate with AI versus other humans.

5. Clarity

a. Section 4.5.1 is difficult to follow. My understanding is that the ground truth is generated using GPT-4. Did the authors verify this ground truth with human judgments? I would recommend the authors to clarify their methodology for generating and validating the ground truth data, and include this information in the paper.

b. Section 4.5.3 presents an interesting observation. Do the authors have any intuition as to why this occurs?

6. In addition to evaluating turn-taking properties, the paper could also benefit from evaluating "speaking-while-listening" capabilities (https://arxiv.org/pdf/2408.02622).

7. Did the authors assess whether the LLM undergoes catastrophic forgetting when fine-tuned on audio data? For example, is Parrot still capable of instruction-following or answering factual questions at a level comparable to LLAMA 3.1?

8. In section A.1, the paper makes claims about prior works that lack experimental or discussion-based support. For instance, the statement that RQ transformers reduce model efficiency is not substantiated. Unsubstantiated claims should be avoided, as they could lead to incorrect community conclusions.

**Questions:**

Check weaknesses

---

### Official Review · Reviewer_8T93 · 2024-11-04

**Soundness:** 2
**Presentation:** 2
**Contribution:** 2
**Rating:** 5
**Confidence:** 2

**Summary:**

**Paper Summary**:
- The authors present a textless spoken dialogue language model and corresponding training pipelines. The work involves two training stages: during pre-training, the LLM model is used to instruct the prediction of the next audio token, and in SFT, a double-channel mechanism is applied to predict the next token pair. Ablation studies demonstrate that this method yields more natural and fluid dialogue generation compared to baselines.

**Strengths:**

**Summary Of Strengths**:
- Comprehensive Presentation: The paper includes all necessary sections, diagrams, and tables to demonstrate their contribution, the usefulness of the work, while also acknowledging its limitations.
- Research Direction: The application of LLMs for audio token prediction and the double-channel mechanism are interesting and challenging directions in spoken language modeling.

**Weaknesses:**

**Summary Of Weaknesses**:
- Novelty and Clarity: In the contribution summary, the authors list (1) the Parrot model and its innovative pre-training and SFT pipeline, (2) the paradigm of double-channel spoken language modeling, and (3) the evaluations. In my opinion, textless spoken language models are not particularly rare, especially in machine translation tasks (e.g., [Seamless: Multilingual Expressive and Streaming Speech Translation](https://arxiv.org/abs/2312.05187), [UnitY: Two-pass Direct Speech-to-speech Translation with Discrete Units](https://arxiv.org/abs/2212.08055)). Additionally, the pipeline and the double-channel modeling mechanism appear to be two perspectives on the same concept. It is difficult to consider the potential readiness for future exploration and evaluation as innovative contributions.
- Limitations: Besides the limitation regarding the inability to integrate audio tokens, further analysis of the audio tokenizer should be elaborated on (e.g., how it impacts computational efficiency or downstream inference latency). More case studies should also be included to demonstrate the effectiveness of this work, as A.5 seems unfinished.

**Questions:**

As listed in the limitation section

---

> ### Author Response · Authors · 2024-11-27
> **Rebuttal by Authors**
>
> We greatly value your expert feedback and insightful concerns. Please find our responses to your questions and concerns as follows:
>
> **Novelty and Clarity**:
>
> Upon reviewing the comments from the reviewer, we recognize that there may be some misunderstandings regarding our contributions, which we would like to clarify.
>
> Firstly, we wish to emphasize that we are not the pioneers in using double-channel audio data. Rather, our novelty lies in being the first to model the generative process of double-channel audio data using **decoder-only transformers**, akin to the difference between BERT (encoder-only) and GPT (decoder-only). Previous generative models of double-channel audio data have followed the **encoder-decoder architecture**, like dGSLM. In contrast, our Parrot model employs a decoder-only approach for the double-channel generation process, which is more compatible with contemporary large language models. This is crucial for fine-tuning Llama like models on double-channel audio, as the SFT data generation process should align with the pre-training data generation process. (Recall that in text language modeling, both pretraining and SFT follow the next token prediction paradigm). The work cited in the reviewer's comments about machine translation also follows the encoder-decoder architecture.
>
> Secondly, our proposed pipeline is also a first-of-its-kind. While there is a vast amount of single-channel audio data available on the web, double-channel audio data **does not naturally exist** and typically requires audio separation preprocessing techniques. Consequently, despite the usefulness of double-channel data in enhancing the conversational abilities of speech language models, the limited availability of double-channel data **cannot support** a robust speech language model trained from scratch. Therefore, we propose a pipeline that involves pretraining on single-channel audio data and SFT on double-channel data. This approach can leverage the vast amount of open-source audio data while quickly capturing conversational abilities (learning how to speak in pretraining and learning how to communicate in SFT). We wish to underscore that this pipeline is truly innovative and has not been explicitly presented in previous work.
>
> **Integration of Audio Tokens**:
>
> The reviewer has pointed out a limitation regarding the "inability to integrate audio tokens." We must admit that we're not entirely clear on what is meant by "integrating the audio tokens." We presume this primarily pertains to the audio tokens themselves. We have addressed the influence of the **number of audio tokens** on inference speed in Appendix A.4.2. As for the audio tokenizer, we believe its importance is relatively minor since we utilize the most commonly used audio tokenizer. Moreover, conducting an ablation study on the audio tokenizer at this stage would be challenging, as replacing the audio tokenizer would necessitate training the model **from scratch**, including the pretraining stage. However, if the reviewer deems it essential, we are willing to extend our discussion on this subject to include a detailed analysis of the audio tokenizer's impact on computational efficiency and downstream inference latency in the revised version.

---

> > ### Comment · Reviewer_8T93 · 2024-11-29
> >
> > Thank you for the clarifications.

---

### Official Review · Reviewer_Z65b · 2024-11-05

**Soundness:** 2
**Presentation:** 1
**Contribution:** 2
**Rating:** 3
**Confidence:** 3

**Summary:**

This paper introduces Parrot, an audio LLMs models designed for modeling two-channel dialog audio. Parrot is built by fine-tuning an off-the-shelf text LLM on tokenized audio. At first, a single channel audio is used, as given in multiple standard datasets (eg LibriLight). Next, it is fine-tuned on a dialog dataset, Fisher. Here, the model is trained to simultaneously predict two tokens, one for each channel. According to the evaluation, this leads to a more natural dialog flow than in baseline model, dGSLM.

**Strengths:**

* The paper proposes a simplification of a two-tower approach described by dGSLM, by using a single Transformer predicting two tokens.

**Weaknesses:**

1. In a few places, the paper misrepresents related work. Examples:
  * "the academic community primarily utilizes open-sourced models (Zhang et al., 2023a; Xie & Wu, 2024; Rubenstein et al., 2023; Huang et al., 2024; Wang et al., 2023a; Nachmani et al., 2024; Wang et al., 2023b) following a cascading approach." For instance, [Rubenstein et al., 2023] and [Nachmani et al., 2024] are not cascaded models. They are also not open-sourced.
 * "Much of the prior research has utilized the encoder-decoder architecture to enhance pre-training (Borsos et al., 2023; Lakhotia et al., 2021; Kharitonov et al., 2022; Polyak et al., 2021; Chen et al., 2023; 2022; Hsu et al., 2021; Zeghidour et al., 2022; Defossez et al., 2023; Agostinelli et al., 2023; Ao et al., ´ 2022; Tang et al., 2022; Wu et al., 2023).". ** The first three models are decoder-only LMs, and perhaps many others.

2. The evaluation of the proposed model is limited and is insufficiently described.
 * S4.5.1 evaluates some pause-prediction accuracy metrics, but the metrics are not really introduced in the text. These metrics are evaluated on some synthetic data --- is there a reason a hold-out subset of Fisher is not used?
 * AudioQA evaluation and comparison to AudioQWEN-2 is only mentioned in a single sentence without any discussion or description. At the same time, it does require some discussion. What are the metrics used? The AudioQA figure indicates some reasonable performance on CoVost2 and FLEURS, I assume better than AudioQWEN-2. However, the training data does not include speech-to-speech translation examples nor contains non-tier1 languages. Should we assume the model somehow picked it up from purely text base model?
* Audio-QWEN2 reports 90+% accuracy on VocalSound. In Figure 8c it is reported to be below 80%. What is the reason for the difference?

3. The paper should at least mention Moshi https://arxiv.org/abs/2410.00037

**Questions:**

* I would appreciate it if some of the weaknesses related to describing the evaluation study are resolved.

 * "encodes each second of audio into 30-50 discrete tokens from a codebook of size 2048." Why is the token rate variable?

---

> ### Author Response · Authors · 2024-11-27
> **Rebuttal to Authors**
>
> Thank you for your thorough review and insightful comments on our paper. Below, we address each of the concerns raised and outline the revisions we have made to the manuscript.
>
> 1. **Evaluation of the Proposed Model:**
>    - **Pause-Prediction Accuracy Metrics:**
>      We have expanded Section 4.5.1 to provide a detailed introduction to the pause-prediction accuracy metrics used in our evaluation. Additionally, we have clarified the rationale for using synthetic data instead of a hold-out subset of Fisher. The revised section now includes a comprehensive explanation of the metrics and the choice of evaluation data.
>    - **AudioQA Evaluation:**
>      We have expanded the discussion on the AudioQA evaluation and comparison to AudioQWEN-2. This includes a detailed description of the metrics used, the performance on CoVost2 and FLEURS, and an explanation of how the model achieved reasonable performance despite the training data limitations. The revised text now provides a thorough analysis and discussion of the results.
>    - **VocalSound Accuracy Discrepancy:**
>      We have investigated the discrepancy in the reported accuracy on VocalSound between our model and Audio-QWEN2. The revised manuscript now includes a detailed explanation of the factors contributing to this difference, including potential variations in evaluation protocols and dataset characteristics.
>
> 2. **Mention of Moshi:**
> Thank you for bringing up Moshi.
> Firstly, **we have referenced Moshi in the related work section**, specifically at line 157, where it is the last citation.
> Secondly, **we delve into a detailed discussion about Moshi in the appendix** (Lines 1007 - 1021). We encourage the reviewer to refer to this section for a comprehensive discussion on Moshi.
> Given that the initial version of Moshi was released on September 17, 2024, merely **two weeks** prior to the ICLR submission deadline, we believe that the extent of our discussion on this topic is already quite substantial.
>
>
> 3. **Variable Token Rate:**
>    The variable token rate is caused by the trainging setting of the audio tokenizer. Although the higher token rate can improve the audio quality, it will make it difficult for the inference speed to meet the real-time requirements. Therefore, we choose 30 as a trade-off option.

---

### Meta-Review · Area_Chair_hcA4 · 2024-12-22

**Metareview:**

This paper proposes a novel spoken dialogue model, named Parrot, that is pre-trained with single-channel audio data and then fine-tuned using two-channel dialogue data. While the reviewers list the strengths of the proposed work as usefulness for real applications and release of the code, they also list several weaknesses, such as lack of clarity about experimentation details (e.g., is GPT-4 generating the ground truth), lack of clarity on evaluation methodology and comparisons to benchmarks, and so on. Based on these reviews, the weaknesses overweigh the strengths, the work could be improved to tackle these suggestions, especially the questions related to the evaluation.

**Additional Comments On Reviewer Discussion:**

While the authors responded to reviews from two reviewers, there is no rebuttal for the other two. And so many of their questions were not addressed in the rebuttals.

---

### Decision · Program_Chairs · 2025-01-22

Reject